# Effects of Structured Expressive Writing on Quality of Life and Perceived Self-Care Self-Efficacy of Breast Cancer Patients Undergoing Chemotherapy in Central China: A Randomized Controlled Trial

**DOI:** 10.3390/healthcare10091762

**Published:** 2022-09-14

**Authors:** Rong Wang, Lu Li, Jing Xu, Zhen-Ting Ding, Jia Qiao, Sharon R. Redding, Yun-Yan Xianyu, Yan-Qiong Ouyang

**Affiliations:** 1Nursing Department of East Campus, Renmin Hospital of Wuhan University, Wuhan 430071, China; 2School of Nursing, Wuhan University, Wuhan 430071, China; 3Global Health of Project HOPE, Omaha, NE 68130, USA

**Keywords:** breast cancer, chemotherapy, expressive writing, quality of life, self-care self-efficacy

## Abstract

Expressive writing is a supportive psychological intervention allowing an individual to disclose and express their deepest thoughts and feelings related to personal traumatic experiences through writing. Previous studies suggested that expressive writing could promote the physical and mental health of cancer patients. The current study was conducted to evaluate the effect of expressive writing based on the theory of cognitive adaptation (TCA) on the quality of life and self-care self-efficacy in patients with breast cancer undergoing chemotherapy. A sample of 82 Chinese women receiving chemotherapy for breast cancer was randomly assigned to an experimental group (four 20 min writing activities focusing on emotional disclosure) or a control group (no writing activities). The quality of life (QoL) and self-care self-efficacy were assessed at baseline, 2 weeks, 4 weeks, and 6 weeks after the intervention, respectively. The sociodemographic characteristics, QoL, and self-care self-efficacy at baseline were comparable between the two groups. Repeated-measures ANOVA revealed significant effects of the time×group (F = 3.65, *p* < 0.05) on the QoL and significant effects of time (F = 4.77, *p* <0.05) on self-care self-efficacy. Compared with the control group, the QoL in the intervention group showed a significant and temporary increase at 2 weeks after the intervention (mean difference = −7.56, *p* < 0.05). As a low-cost and easily delivered psychological intervention, expressive writing is recommended to reduce stress when there is a lack of available emotional support.

## 1. Introduction 

The International Agency for Research on Cancer (IARC) showed that there were approximately 19.3 million new cancer cases and 10.0 million cancer deaths worldwide in 2020 [1]. Among them, breast cancer has become the most common cancer in the world, with approximately 2.26 million (11.7%) new cases and 680,000 deaths in 2020 [2]. Over the past 15 years, the age-standardized incidence rate of breast cancer in China has increased by approximately 6.9% per year in rural areas and 2.6% per year in urban areas according to the National Central Cancer Registry (NCCR) of China [3]. Moreover, it will grow by 15.6% in 2030 in China, which is even more significant [3]. 

In addition to the distress and psychological shock caused by cancer, the aggressive nature of some treatments has resulted in unexpected physical and psychological side effects, although improved detection and treatment of breast cancer has led to a steep rise in survival rates. As a mainstay of cancer treatment, chemotherapy may cause fatigue, nausea, vomiting, anorexia, sleep disorders, and other adverse effects [4,5]. With ongoing treatment, patients may develop psychological disorders due to increased concern about their disease prognosis [6]. These negative and subjective parameters related to health status may persist into post-treatment and negatively affect the quality of life of patients with breast cancer [7]. The quality of life (QoL), specifically referring to breast cancer patients’ perception of their own physical, mental, and social health that is influenced by diagnosis, treatment, post-treatment, and survivorship, is an important predictor of mortality and recurrence in breast cancer patients [8]. Asian breast cancer patients including Chinese patients with comorbidities and those who were treated with chemotherapy had poorer health-related QOL [9]. Self-care self-efficacy is an indicator of a person’s confidence in performing relevant behaviors in a self-care situation and is related to the positive physical and mental health of cancer patients [10]. In order to promote patients’ self-care behaviors and improve their psychosocial and functional status outcomes, their self-care self-efficacy should be bolstered first [10,11]. Self-care self-efficacy has a major impact on the quality of life and psychological well-being.

As a form of emotional expression, expressive writing was proposed in 1986 by Pennebaker and Beall [12]. Expressive writing is a low-cost, easily delivered, and manageable psychological intervention that allows an individual to disclose and express their deepest thoughts and feelings related to personal traumatic experiences through writing [13,14]. Although previous studies have demonstrated the effectiveness of expressive writing in patients with dementia and irritable bowel syndrome, including improving patients’ QoL and self-efficacy and reducing healthcare utilization, it remains to be tested in patients with breast cancer [15,16]. In China, a considerable number of patients with breast cancer who are receiving chemotherapy are entering the period of rehabilitation. Expressive writing may assist them to make psychological adjustments, which will play a positive role in their return to family and social life. Furthermore, expressive writing, which does not require expensive equipment or specialized therapists, is regarded as a good way to help patients cope with psychological challenges from cancer [14]. Currently, in China, there are few studies on the psychological status of patients with breast cancer undergoing chemotherapy and the impact of expressive writing on these patients. Thus, the current study was conducted to evaluate the effect of expressive writing on the QoL and self-care self-efficacy in patients with breast cancer undergoing chemotherapy.

## 2. Methods 

### 2.1. Study Design and Participants

A single-blinded randomized controlled trial was conducted between December 2021 and February 2022 at two departments of a university-affiliated hospital in central China. Ethics approvals were granted from the ethics committee of the tertiary hospital and the authors’ institution. The participants were recruited when commencing their chemotherapy cycle either in the Breast Surgery Department or Chemotherapy Department. For the purposes of the current study and referring to the study of Lu et al. [17], α, 1-β, and effective size were set at 0.05, 0.8, and 0.2, respectively. G*Power version 3.0.5 was used for sample size estimation; the final sample size needed was 82, 41 people in each group, and the loss rate was 15%. Inclusion criteria were women (a) diagnosed with breast cancer having completed surgery; (b) currently receiving chemotherapy; (c) who were able to read, write, and speak Chinese; (d) aged 18 years or older; and (e) who were able to independently complete a 20 min writing exercise. Exclusion criteria were women (a) undergoing any form of psychotherapy; (b) with a habit of keeping diaries; and (c) having a diagnosis of mental illness or severe cognitive impairment. Potential participants were screened based on their health records, and eligible patients were asked if they agreed to participate in the study. After determining their eligibility, the participants were randomized into an expressive writing group (intervention group, IG) or a treatment-as-usual group (control group, CG) using the block randomization method. A random sequence was computer-generated, and the group assignment was carried out by a researcher using an opaque sealed envelope.

### 2.2. Intervention Development 

The theory of cognitive adaptation (TCA) was used to guide the current intervention because of its emphasis on cognitive adjustment to a threatening life event involving a search for meaning in the experience and an attempt to restore one’s sense of control and positive self-view [18]. The three themes identified by TCA were meaning, mastery, and self-enhancement. After literature review, group discussion, and expert consultation, an expressive writing plan for patients with breast cancer undergoing chemotherapy was determined by a research group composed of an oncologist, surgeon expert in breast cancer, postgraduate nursing student, professor of gynecological nursing, and clinical psychologist. The theme and focus of each writing session are shown in Table 1. 

### 2.3. Intervention Procedure

Instructions for the writing activity and paper were enclosed in envelopes and sent in advance to the participants in the IG. They were asked to independently write for 20 min at a time without talking to others and complete one writing assignment in a week (i.e., emotional disclosure, cognitive appraisal, benefit finding, and looking to the future) on four consecutive days. If a participant was discharged from the hospital before the completion of the expressive writing, that individual could finish the writing exercise at home. The completed assignment was then mailed to the researcher. Any participant experiencing distress following the activity could be referred to the clinical psychologist for consultation if necessary. The participants in both groups were provided materials about breast cancer recovery.

The sociodemographic characteristics of the participants in both groups were collected at baseline (T0), and the QoL and perceived self-care self-efficacy were assessed at baseline (T0), 2 weeks (T1), 4 weeks (T2), and 6 weeks (T3) after the intervention, respectively. A 2-week period was chosen to allow the participants to fully process the ideas inspired by writing without diluting the effect of the intervention [19].

### 2.4. Measures

#### 2.4.1. Sociodemographic Characteristics

Sociodemographic characteristics were collected using a researcher-designed questionnaire including age, religion, marital status, educational level, income, stage of breast cancer, type of surgery, and times of chemotherapy treatments.

#### 2.4.2. The Quality of Life Instruments for Patients with Breast Cancer (QLICP-BR)

The participants’ QoL level was assessed using the QLICP-BR, which is a 5-point Likert scale, with scores ranging from 1 ("not at all") to 5 ("very much") [20]. A higher score indicates a better quality of life. This scale was developed by Wan et al., and it consists of 39 items including the Quality of Life Instruments for Cancer Patients—General Module (QLICP-GM) and seven items specific to breast cancer. The QLICP-GM includes four domains: physical function (seven items), psychologic function (12 items), social function (six items), and common symptoms and side effects (seven items) [21]. Cronbach’s α of the total scale and four domains were 0.81, 0.84, 0.73, 0.89, and 0.69, respectively [22]. In the pilot study, Cronbach’s α of the total scale was 0.88, and those of the four domains were 0.72, 0.86, 0.77, and 0.71, respectively, suggesting that the scale was in the acceptable range.

#### 2.4.3. The Strategies Used by People to Promote Health (SUPPH) Scale

The Strategies Used by People to Promote Health (SUPPH) scale was developed to measure self-care self-efficacy in the cancer patient population. Each item of the SUPPH is rated on a 5-point Likert scale of confidence from 1 ("very little") to 5 ("quite a lot"). Higher scores indicate more positive perceptions of self-efficacy. The original SUPPH scale was developed by Lev and Owen, and the initial version included 29 items, divided into three domains: positive attitude (16 items), stress reduction (10 items), and making decisions (three items). Yuan et al. introduced it to China and made culture adaptations [23,24]. There were 28 items in the Chinese version of SUPPH (C-SUPPH), including positive attitude (16 items), stress reduction (nine items), and self-decision (three items). Cronbach’s α of the C-SUPPH and three domains were 0.93, 0.92, 0.89, and 0.83, respectively [24]. In the pilot study, Cronbach’s α of the total scale was 0.85, and those of the three domains were 0.89, 0.93, and 0.87, respectively, suggesting that the scale evidenced adequate internal consistency.

### 2.5. Statistical Analysis

Numerical variables were described by mean and standard deviation (mean ± SD) and categorical variables by percentage distribution. The chi-square test was used for the comparison of categorical variables and *t*-test for quantitative variables. Repeated-measures ANOVA and simple effect analysis were used to investigate the interaction effects of time × group and the main effects of time and group on the QoL and self-care self-efficacy. Pearson’s correlation coefficient was used to examine the relationship between the QoL and self-care self-efficacy. All tests were two-sided, and the probability of a type I error was set at *p* < 0.05. Analyses were performed using SPSS version 23.0 (IBM Corp, New York, NY, USA).

### 2.6. Ethical Considerations

This study was approved by the ethics committees of the tertiary hospital and the authors’ institution (ethics approval number: 2020YF0083). Moreover, all participants signed the written informed consent agreement after receiving information about the study. The data were used only for the purpose of this study, and the participants were allowed to anonymously complete the questionnaire and scale.

## 3. Results

### 3.1. Sample Characteristics

In total, 82 breast cancer patients receiving chemotherapy were randomized into the IG or CG. During the intervention, five participants in the IG dropped out due to physical limitations (n = 3), refusal to continue (n = 1), and discontinuation of treatment (n = 1), whereas two participants in the CG dropped out due to refusal to continue (see Figure 1). The final sample size included in the analysis was 75 (IG: n = 36; CG: n = 39). The average age of participants in the total sample was 47.99 years (SD = 9.23). Most of the women were married (88.0%) and in stage II of breast cancer (72.0%). Table 2 shows no significant differences in the sociodemographic characteristics between the IG and CG by *t*-test or chi-square test (*p* > 0.05), indicating the success of randomization.

### 3.2. Effects on QoL and Perceived Self-Care Self-Efficacy

A *t*-test of two independent samples was used to compare the scores of the participants in both groups of the QLICP-BR, SUPPH, and each dimension at baseline. The results showed no significant differences (*p* > 0.05), indicating that the QoL and perceived self-care self-efficacy of the two groups at baseline were comparable. The scores of the participants’ QLICP-BR and SUPPH at different time points are shown in Table 3.

Relating to the QoL, Mauchly’s test of sphericity showed that the data did not meet the assumption of sphericity (*p* < 0.05), so the results of multivariate tests were described. Repeated-measures ANOVA revealed significant interaction effect between the time × groups (*F* = 3.65, *p* = 0.016), suggesting that the trends of the intervention effect over time were different between the two groups (see Table 3). The results of further simple effect analysis indicated no significant differences in the scores of the QLICP-BR between the IG and CG at any time (*p* > 0.05). Regarding the IG, the QoL at T1 after the intervention was significantly higher than at baseline, T2, and T3 after the intervention (*p* < 0.05), and there were no significant differences at baseline, T2, and T3 after the intervention (*p* > 0.05). Descriptively, the QoL was observed to increase from T0 to T1 (mean difference = −7.56, *p*= 0.003) and to decrease from T1 to T3 in the intervention group (see Figure 2). For the control group, although the QoL improved from T1 to T2 after the intervention, the change was not significant (mean difference = −3.58, *p* > 0.05),* means *p* < 0.05.

Similarly, multivariate test results were presented because the data on self-care self-efficacy did not meet the assumption of sphericity (*p* < 0.05). A nonsignificant interaction between time and group (*F* = 0.11, *p* = 0.954) and a nonsignificant group effect on self-care self-efficacy were observed (*F* = 1.00, *p* = 0.320). However, there was a significant main effect of time (*F* = 4.77, *p* = 0.004) (see Table 3). Further analyses showed significant differences in self-care self-efficacy between baseline and T1 (mean difference = 9.20, *p* < 0.001), T2 (mean difference = 6.69, *p* = 0.015), and T3 (mean difference = 6.24, *p* = 0.021) after the intervention. The scores on self-care self-efficacy at T1 after the intervention were significantly lower than those at baseline, and there was no significant increase from T1 to T3 after the intervention (see Figure 3).

## 4. Discussion

This study explored the effectiveness of an expressive writing intervention among patients with breast cancer undergoing chemotherapy in central China. The findings showed that the QoL was significantly higher at 2 weeks than at other timepoints after the intervention. Cognitive adaptation is an important psychological coping resource for the individual managing traumatic experiences such as cancer according to the cognitive adaptation theory. Previous studies showed that expressive writing enhanced self-affirmation, alleviated cancer-related stress, and positively affected functional health status [25]. However, in the current study, expressive writing showed only a temporary benefit immediately after an intervention in terms of the QoL and no significant improvement in self-care self-efficacy. Regarding breast cancer patients, ‘‘self-care’’ mainly refers to maintaining a positive attitude, reducing stress, and making appropriate decisions during long-term chemotherapy. A sense of “self-efficacy” in cancer patients is likely to be positively associated with patients’ behavioral efforts to perform challenging self-care tasks and to expecting positive outcomes [24].

Prior research has shown that individuals who actively suppress their emotions, thoughts, and actions for an extended period of time may turn them into a chronic stressor that severely impairs physical and mental functioning. Through expressive writing, participants focus on different aspects of traumatic or illness experiences, which not only makes them more accustomed to a stressful event and its influence but also facilitates a more adaptive reorganization of feelings and thoughts, leading to increased self-efficacy [26]. Cognitive adaptation theory provides a new theoretical perspective for explaining how people integrate internal resources, promote individual adaptation, and restore and develop psychological functions in extremely stressful events or situations. An empirical study of cognitive adaptation theory in cancer patients suggested that a cognitive adaptation index, including optimism, self-esteem, and control, was associated with the QoL of advanced cancer patients [26]. Based on the above study, this current study applied cognitive adaptation theory to expressive writing of breast cancer patients and found immediate and significant improvement in the QoL.

Results of a meta-analysis showed that expressive writing had a significant effect on the QoL of cancer patients after subgroup analysis, but its effect on patients with any type of cancers was uncertain [27]. Women with breast cancer after surgical treatment may still receive six to eight cycles of chemotherapy with the TAC regimen (including doxorubicin, docetaxel, and cyclophosphamide) [28]. Extant research suggested that the physical, social and family, psychological, and functional aspects of the quality of life significantly differ by chemotherapy cycles [29]. Therefore, it is suggested that clinicians should provide more attention and assistance to breast cancer patients in the fourth and fifth cycles to prevent negative outcomes due to side effects. The effect of expressive writing may vary depending on the stage and treatment of cancer, and patients with newly diagnosed breast cancer may benefit from cancer-fact writing and emotional disclosure, whereas those with advanced or recurrent cancer may not [24,27]. Obviously, the differences in chemotherapy cycles and the individual’s responses were critical to the health outcomes of cancer patients.

The organization and presentation of the intervention may be one of the vital reasons for the ineffectiveness of expressive writing, although there are currently no standards for the number, duration, and frequency of interventions. Pennebaker and Beall’s original standardized expressive writing intervention consisted of 4 days of 20 min of writing about feelings and thoughts of traumatic experiences [30]. However, compared with a healthy population, cancer patients have more complex traumatic experiences and take longer to reveal their innermost thoughts and feelings, so 20 min may not be sufficient [27]. The meta-analysis results of Reinhold et al. indicated that the frequency and focus of expressive writing were significantly related to the intervention effect, and the effect was greater when there were more times of writing and specific writing topics [31].

There are individual differences in the effectiveness of expressive writing (e.g., diagnosis time and emotional support). Individuals may be differently affected by the same diagnosis, and the resilience to stressful events can vary from one person to another. Meanwhile, research has shown that expressive writing contributes more to the QoL of patients who receive social support [32]. Moreover, an expressive writing intervention at home might not be as effective as in a clinical setting. Although emotional disclosure was more natural in a family setting, it might increase the likelihood of non-adherence and surrounding disturbance [25].

Some limitations should be acknowledged. The effects of individual differences (e.g., diagnosis time and emotional support) on the QoL and self-care self-efficacy were not considered in this study. Previous psychological interventions have been shown to be more effective for people in extreme distress. Moreover, writing locations were not uniform. Participants writing at home might dilute the effect of expressive writing interventions. Future research should develop an expressive writing intervention program with uniform writing location, prolonged duration, and diverse topics following a psychologist’s suggestions, and giving priority to patients with breast cancer who need psychological intervention.

## 5. Conclusions

In general, structured expressive writing did not improve the QoL and self-care self-efficacy of patients with breast cancer undergoing chemotherapy. However, one generalization is worth mentioning.

As a low-cost and easily delivered psychological intervention, it is recommended to reduce stress when there is a lack of available emotional support. More studies are needed in the future to explore the effectiveness of expressive writing in patients at different stages of cancer and treatment.

## Figures and Tables

**Figure 1 healthcare-10-01762-f001:**
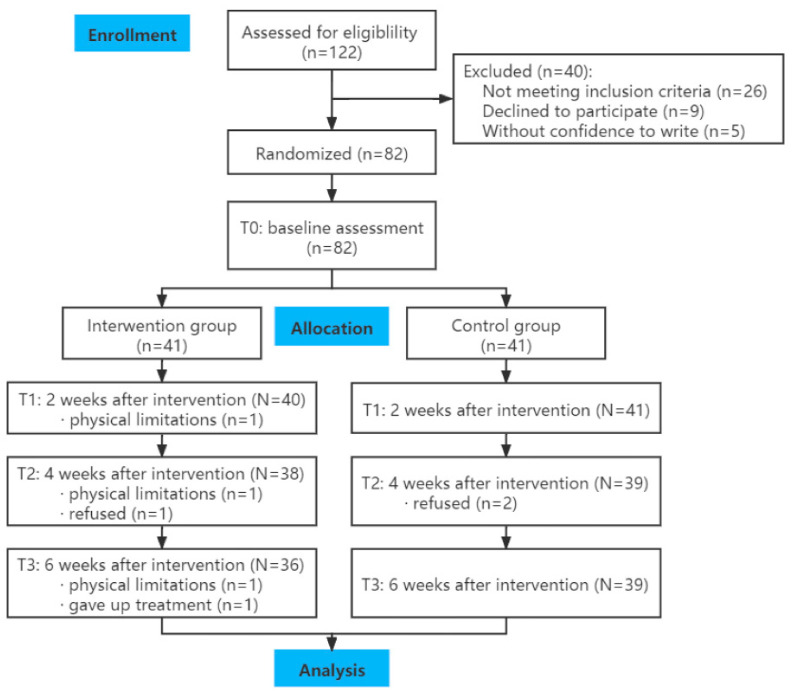
CONSORT Diagram.

**Figure 2 healthcare-10-01762-f002:**
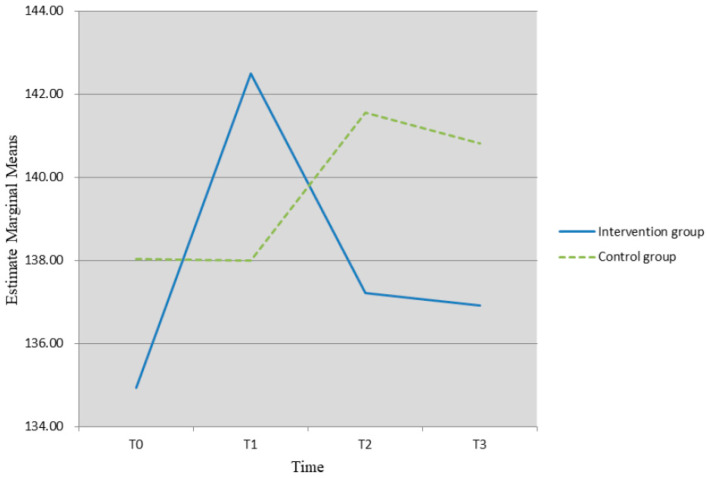
Change of QoL from T0 toT3 in intervention and control groups.

**Figure 3 healthcare-10-01762-f003:**
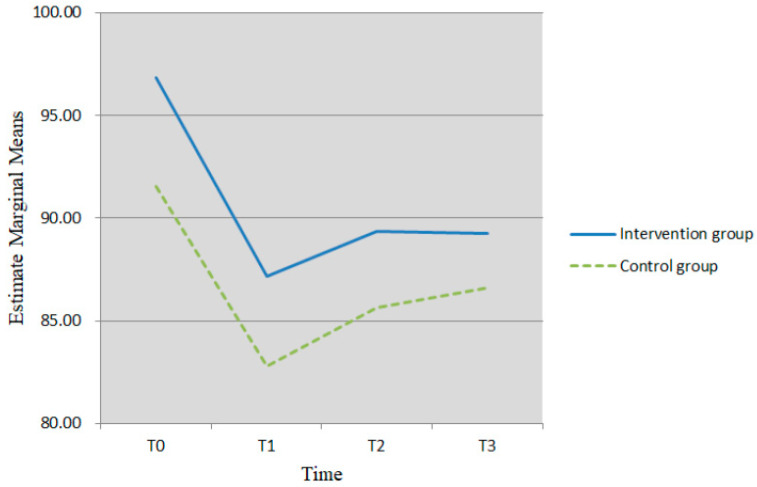
Change of self-care self-efficacy from T0 toT3 in intervention and control groups.

**Table 1 healthcare-10-01762-t001:** Writing themes.

Session	Theme	Focus of Content
I	Emotional disclosure	1. Describe your deepest thoughts and feelings about the experience of breast cancer. 2. Describe detailed sensory experiences of breast cancer.
II	Cognitive appraisal	1. What might be the cause for your illness? 2. What does breast cancer mean to you? 3. What does your life mean now? 4. How do you treat this disease to make it more meaningful? 5. Write advice on how to deal with breast cancer and then apply it to yourself.
III	Benefit finding	1. What benefits have you perceived? 2. What challenges have you overcome? 3. Describe the changes in your outlooks on life or priorities. 4. Make a comparison with other patients and try to find positive aspects. 5. What is the effect of attitude on disease in your mind?
IV	Looking to the future	1. What is your coping strategy in the face of breast cancer? 2. How do you make efforts? 3. Write down the experience you want to share with others. They can be your relatives, friends, or other patients. 4. In your opinion, what will change in the future?

**Table 2 healthcare-10-01762-t002:** Comparison of sociodemographic characteristics between IG and CG.

	IG (n = 36)	CG (n = 39)	*p*
Age (years), (M ± SD)	48.08 ± 9.71	47.90 ± 9.02	0.932
Religious belief, n (%)			0.919
Yes	3 (8.3)	3 (7.7)	
No	33 (91.7)	36 (92.3)	
Marital status, n (%)			0.629
Single	5 (13.9)	4 (10.3)	
Married	31 (86.1)	35 (89.7)	
Education, n (%)			0.547
Middle school or below	14 (38.9)	16 (41.0)	
High school or junior college	14 (38.9)	18 (46.2)	
College or above	8 (22.2)	5 (12.8)	
Monthly income (in CNY), n (%)			0.210
≤3000	20 (55.6)	29 (74.4)	
3001–6000	11 (30.6)	6 (15.4)	
>6000	5 (13.9)	4 (10.2)	
Stage of breast cancer, n (%)			0.502
I	0	1 (2.6)	
II	25 (69.4)	29 (74.4)	
III	11 (30.6)	9 (23.1)	
Type of breast surgery received, n (%)			0.499
Mastectomy	10 (27.8)	9 (23.1)	
Breast conservation treatment	25 (69.4)	30 (76.9)	
Breast reconstruction	1 (2.8)	0	
Chemotherapy treatment (times)	4.94 ± 4.12	5.23 ± 7.00	0.831

**Table 3 healthcare-10-01762-t003:** Scores of QoL and self-care self-efficacy, and results of repeated-measures ANOVA.

Measure	Time	IG (n = 36)	CG (n = 39)	Group	Time	Group × Time
		Mean (SD)	Mean (SD)	*F*	*p*	*F*	*p*	*F*	*p*
QLICP-BR				0.27	0.606	1.71	0.173	3.65	0.016 *
	T0	134.94(17.11)	138.04(18.81)						
	T1	142.50(15.55)	137.99(19.71)						
	T2	137.21(17.60)	141.56(16.52)						
	T3	136.92(14.76)	140.82(16.04)						
SUPPH				1.00	0.320	4.77	0.004 *	0.11	0.954
	T0	96.83(25.25)	91.54(24.14)						
	T1	87.19(20.48)	82.77(18.01)						
	T2	89.36(18.74)	85.64(21.16)						
	T3	89.28(22.10)	86.62(16.93)

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
