# Peer review of "Effects of Structured Expressive Writing on Quality of Life and Perceived Self-Care Self-Efficacy of Breast Cancer Patients Undergoing Chemotherapy in Central China: A Randomized Controlled Trial"

_healthcare, 2022, doi:10.3390/healthcare10091762_

Round 1

Reviewer 1 Report

1- It is necessary to explain the definition of expressive writing in the abstract.

2- It is better to describe the quality of life of patients in the introduction according to past studies.

3- It is necessary to write the place of research in the title

4- It is necessary to explain more about the intervention methods.

5- The tools used in the research should be explained more

6-It is necessary to discuss the results of other people's studies more

Author Response

Reviewer 1

Thank you very much for your kind and comprehensive instructions! We have learned a lot from you.

Reviewer 1:

1- It is necessary to explain the definition of expressive writing in the abstract.

Authors:

1-Indeed, it is necessary that the definition of expressive writing should be added in the Abstract  section. We have marked this in red font.

Reviewer 1:

2- It is better to describe the quality of life of patients in the introduction according to past studies.

Authors:

2-We have revised the Introduction section following your advice.

Reviewer 1:

3- It is necessary to write the place of research in the title.

Authors:

3-Thank you and we have done so.

Reviewer 1:

4- It is necessary to explain more about the intervention methods.

Authors:

4- Yes, your suggestion is very precious for us! Thank you very much and we have added more to complement the entire intervention program.

Reviewer 1:

5- The tools used in the research should be explained more.

Authors:

5-Your preciseness offers a deep impression. We are happy to follow  your instruction.

Reviewer 1:

6-It is necessary to discuss the results of other people's studies more.

Authors:

6-Yes, we have revised the related paragraph.

Reviewer 2 Report

1, similar articles have been published: expressive writing[ti] AND breast cancer[ti], the innovation is compromised. 

Author Response

Reviewer 2:

1, similar articles have been published: expressive writing[ti] AND breast cancer[ti], the innovation is compromised.

Authors:

1, Your comment is crucial for us, and we appreciate your frank suggestion and  kind assistance very much!  We have revised this manuscript as best as we can. We will place more effort on the innovation in our further study. Thank you again!

Reviewer 3 Report

The autors present a randomized controlled trial on the effects of structured expressive writing on quality of life and perceived self-care self-efficacy of breast cancer patients undergoing chemotherapy. 82 patients were analyzed in ragard to QoL and self-efficacy.

The manuscript is well written, has a sound scientific approach, and the data is presented in a transparent way. The citations are adequate.

Here are some minor suggestions for further improvement:

1. p1, keywords: spelling cancer

2. p1: Introduction first sentence: would data not be better than statistics in this case?

3. The introduction is well written, but it appears a little bit unfocused and superficial. Wouldn't it be better not to write so much about breast cancer incidence in general, but to raise the authors attention to OoL in breast cancer and maybe to give an overview on established and verified treatments to improve QoL during treatment?

4. p2/methods: "The trial was retrospectively registered in the ..." is not a sign of quality for a propective randomized trial. Please clearly state the dates of trial registration, of the ethics commitee approval and the inclusion of the first patient in the manuscript.

5. Sample size calculation and radomisation procedure appear adequate.

6. p3, last line: shift subheading to the next page

7. In the whole manuscript there is no word at all on the effect of radiotherapy on QoL in this subgroup of patients. Did the patients receive radiotherapy after the end of this examination? please specify in the manuscript.

8. p5, line4: two-sided?      

9: Figure 1: this is usually called a CONSORT diagram

10: Tables 2 and 3: the forth column is not helpful for the reader. Please delete. Non-significant p-values of this kind: 0.876 are not helpful and reflect a pseudoprecision. Please change to: n.s. (not significant). 

11. p10, line3: please state how long the chemotherapy usually lasted, did the examination time points of the study cover the whole period of chemotherapy?

12. It is stated that expressive writing at home was done by a part of the study population and it is discussed if this was not so effective regarding the endpoints. is there any data on compliance and study intervention adherence for the IG group. Please speciufy in the manuscript.

13: Conclusion: Please delete "it is expected" in the second sentence, or better the whole sentence. This is not covered by your own results and cost-effectiveness as well as easy delivery alone are no arguments to establish a medical intervention.

Author Response

Reviewer 3:

  1. p1, keywords: spelling cancer

Authors:

  1. Thank you very much for your careful review. The spelling has been modified.

Reviewer 3:

  1. p1: Introduction first sentence: would data not be better than statistics in this case?

Authors:

2.Yes, we totally agreed with your suggestions. We have revised the first sentence with red font.

Reviewer 3:

  1. The introduction is well written, but it appears a little bit unfocused and superficial. Wouldn't it be better not to write so much about breast cancer incidence in general, but to raise the authors attention to OoL in breast cancer and maybe to give an overview on established and verified treatments to improve QoL during treatment?

Authors:

3.We absolutely agreed with your opinions. We have revised the Introduction section, updated some information about  QoL in breast cancer patients undergoing chemotherapy.

Reviewer 3:

  1. p2/methods: "The trial was retrospectively registered in the ..." is not a sign of quality for a propective randomized trial. Please clearly state the dates of trial registration, of the ethics commitee approval and the inclusion of the first patient in the manuscript.

Authors:

4.We appreciate your understandin very much! We are aware that a retrospective registration is not a sign of quality in a single-blinded randomized controlled trial. For this regard, we consulted with the academic officer of the Chinese Clinical Trial Register and asked for their assistance.Fortunately, we got their permission for registration. We are very sorry for the inconvenience. This trail was reviewed by the ethics committees of the tertiary hospital and the authors’ institution, and the ethics approvals were guaranteed. The pilot study involving 10 participants was initiated upon getting the ethics approvals. After determining the feasibility and acceptability, the participants were recruited when commencing chemotherapy.

Reviewer 3:

  1. Sample size calculation and radomisation procedure appear adequate.

Authors:

5.Thank you very much for your encouragement! We will make greater effort in the future.

Reviewer 3:

  1. p3, last line: shift subheading to the next page

Authors:

6.Thank you for your carefulness. We have followed your instrction.

Reviewer 3:

  1. In the whole manuscript there is no word at all on the effect of radiotherapy on QoL in this subgroup of patients. Did the patients receive radiotherapy after the end of this examination? please specify in the manuscript.

Authors:

7.Thank you for your query. Actually, in the whole manuscript there is no word at all on the effect of radiotherapy on QoL in this subgroup of patients. The participants didn’t receive radiotherapy, but were undergoing chemotherapy.

Reviewer 3:

  1. p5, line4: two-sided?   

Authors:

8.Yes, two-sided.

Reviewer 3:

9: Figure 1: this is usually called a CONSORT diagram

Response

  1. Yes. CONSORT diagram.

Reviewer 3:

10: Tables 2 and 3: the forth column is not helpful for the reader. Please delete. Non-significant p-values of this kind: 0.876 are not helpful and reflect a pseudoprecision. Please change to: n.s. (not significant). 

Authors:

10.Yes, you are right. Table 2 was shown clearly and concisely when deleting the fourth column. As  for the fifth column in Table 2, it is better to be retained referring to the article published in Healthcare (see the reference).

As far as Table 3 is concerned, since the values of P and F  presented different meaning, it will be confusing if this is deleted. We hope you understand our position.

Reference:  Laura P,Giulia S,Renzo P,et al. Effectiveness of Expressive Writing in Kidney Transplanted Patients: A Randomized Controlled Trial Study.Healthcare 2022, 10(8), 1559; https://doi.org/10.3390/healthcare10081559

Reviewer 3:

  1. p10, line3: please state how long the chemotherapy usually lasted, did the examination time points of the study cover the whole period of chemotherapy?

Authors:

11.The chemotherapy usually lasted for 6-8 cycles after surgical treatment (P10.line 2 in red font), as for the participants in the current study, some were covered the entire period and some not.

Reviewer 3:

  1. It is stated that expressive writing at home was done by a part of the study population and it is discussed if this was not so effective regarding the endpoints. is there any data on compliance and study intervention adherence for the IG group. Please speciufy in the manuscript.

Authors:

  1. We just speculated that participants writing at home might be influenced by the environmental stimulus, diluting the effect of expressive writing interventions. We are sorry that we didn’t have any data on compliance and intervention adherence.

Reviewer 3:

13: Conclusion: Please delete "it is expected" in the second sentence, or better the whole sentence. This is not covered by your own results and cost-effectiveness as well as easy delivery alone are no arguments to establish a medical intervention.

Authors:

13.Thank you very much. We have deleted this sentence following your advice.

Round 2

Reviewer 2 Report

Reject

Response: 

We appreciate your last round suggestion and kind assistance very much!  We have tried our best to revise this manuscript. Thank you again!